# Inhibition of FGFR Signaling by Targeting FGF/FGFR Extracellular Interactions: Towards the Comprehension of the Molecular Mechanism through NMR Approaches

**DOI:** 10.3390/ijms231810860

**Published:** 2022-09-17

**Authors:** Katiuscia Pagano, Elisa Longhi, Henriette Molinari, Giulia Taraboletti, Laura Ragona

**Affiliations:** 1Istituto di Scienze e Tecnologie Chimiche “Giulio Natta” (SCITEC), via Corti 12, 20133 Milano, Italy; 2Laboratory of Tumour Microenvironment, Department of Oncology, Istituto di Ricerche Farmacologiche Mario Negri IRCCS, 24126 Bergamo, Italy

**Keywords:** NMR, DOSY, fibroblast growth factor, rosmarinic acid, resveratrol, dobesilate, allosteric inhibitors

## Abstract

NMR-based approaches play a pivotal role in providing insight into molecular recognition mechanisms, affording the required atomic-level description and enabling the identification of promising inhibitors of protein–protein interactions. The aberrant activation of the fibroblast growth factor 2 (FGF2)/fibroblast growth factor receptor (FGFR) signaling pathway drives several pathologies, including cancer development, metastasis formation, resistance to therapy, angiogenesis-driven pathologies, vascular diseases, and viral infections. Most FGFR inhibitors targeting the intracellular ATP binding pocket of FGFR have adverse effects, such as limited specificity and relevant toxicity. A viable alternative is represented by targeting the FGF/FGFR extracellular interactions. We previously identified a few small-molecule inhibitors acting extracellularly, targeting FGFR or FGF. We have now built a small library of natural and synthetic molecules that potentially act as inhibitors of FGF2/FGFR interactions to improve our understanding of the molecular mechanisms of inhibitory activity. Here, we provide a comparative analysis of the interaction mode of small molecules with the FGF2/FGFR complex and the single protein domains. DOSY and residue-level NMR analysis afforded insights into the capability of the potential inhibitors to destabilize complex formation, highlighting different mechanisms of inhibition of FGF2-induced cell proliferation.

## 1. Introduction

Fibroblast growth factor (FGF2)/fibroblast growth factor receptor (FGFR) signaling represents a crucial element of progression in several tumors driven by aberrant FGFR activation [1,2]. FGF/FGFR interactions play relevant roles in the organization of the tumor microenvironment and the regulation of tumor–stroma interactions, including immune evasion. The FGF/FGFR pathway has been recently recognized as a mediator of tumor resistance to therapies [3]. Several inhibitors of this pathway have been developed, including erdafitinib and pemigatinib, which have been approved for metastatic urothelial carcinoma and advanced or metastatic cholangiocarcinoma with FGFR alterations, respectively [4,5,6]. Besides cancer, the FGF/FGFR axis acts as a pivotal player in other pathologies, including angiogenesis-driven and vascular diseases, expanding the potential range of therapeutic applications for inhibitors of this pathway [7].

FGF2 is the prototype and most studied member of the FGF family. FGF interactions with the FGFR extracellular D2 and D3 domains and heparin induce the auto-phosphorylation of the FGFR intracellular tyrosine kinase (TK) domain, leading to the activation of FGFR signal transduction pathways (Figure 1A) [8,9,10].

Most FGFR inhibitors developed to date, such as erdafitinib and pemigatinib, target the ATP binding pocket of the FGFR intracellular portion (Figure 1A) [12]. However, limited specificity, relevant toxicity, and the occurrence of intrinsic and acquired resistance limit the efficacy of these drugs, even in RTK-addicted cancers [13]. Inhibition of the FGF/FGFR system through molecules acting at the extracellular level offers new pharmacological opportunities [14,15,16,17]. Targeting allosteric regulatory sites is a potentially powerful strategy [15], despite only one molecule having been tested in a clinical trial to date, targeting the extracellular portion of the FGFR receptor [18,19].

Protein–protein interactions play crucial and subtle roles in many biological processes, and the modulation of their association by specific molecules represents a promising therapeutic opportunity. NMR spectroscopy techniques provide atomic resolution information on the binding interfaces, intermolecular affinity, and binding-induced conformational changes in protein–protein complexes formed in solution [20], thus playing a determinant role in the identification of potential inhibitors of protein–protein interactions.

Previous NMR studies support the identification of a few leads, either targeting FGF2 [21,22,23] or the FGFR-D2 domain [24], able to interfere with the FGF2/FGFR system and inhibit the subsequent signaling cascade. Specifically, the synthetic compound SM27 (Figure 1), mimicking the endogenous inhibitor of angiogenesis thrombospondin-1, engages FGF2 at the heparin binding site and induces long-range dynamic perturbations along the FGF2/FGFR interface regions [21,25] (Figure 1B). The functional consequence of SM27 binding is an impaired FGF2 interaction with FGFR and heparan sulfate proteoglycans, as demonstrated by SPR and cell-based binding assays [21].

To capitalize on previous results, we extended the mechanistic interaction studies to a small library of natural and synthetic compounds, namely resveratrol, toluquinol, and calcium dobesilate, which were selected based on their vaunted potential as inhibitors of the FGF2/FGFR axis (*vide infra*). SM27 is included in the library, as additional experiments are required to investigate whether it can target the receptor domain (Figure 1).

Resveratrol (RES) is a polyphenolic compound reported to inhibit the phosphorylation of mitogen-activated kinase isoforms (MAPKp44/MAPKp42) induced by FGF2 in proliferating endothelial cells in a dose-dependent manner [27]. It belongs to the stilbenoid class of naturally occurring phytochemicals known to display prominent antiangiogenic effects in preclinical models of tumor angiogenesis [26,28,29,30]. Toluquinol (TOL) is a marine fungus metabolite reported to affect the FGF2-induced phosphorylation of Akt in BAECs cells and FGF2-mediated cell invasion in the Matrigel plug [31]. 2,5-dihydroxybenzene sulfonate (dobesilate, DOBE) is an anti-inflammatory and antiviral synthetic compound belonging to a group of chemical leads derived from gentisic acid and identified as an effective anti-FGF inhibitor [32,33].

Cellular tests were performed to validate their biological activity as modulators of the FGF2/FGFR pathway. NMR solution studies were employed to assess the capability of the small molecules to target the FGF2/FGFR complex and highlight any preferential binding with the single proteins.

NMR diffusion experiments enabled the evaluation of the fraction of protein–protein association in solution and quantification of the effect of the small molecules on the equilibrium between complexed and free protein species. DOSY thus highlighted the capability of the different ligands to dissociate the FGF2/D2 complex. Finally, a residue level analysis of protein perturbations suggested the molecular mechanism employed by the studied molecules to act as FGFR signaling inhibitors. The reported comparative approach serves the general purpose of identifying the most active inhibitors for a more in-depth molecular and cellular investigation. The knowledge provided by the NMR analyses of the molecular mechanism of action of the small molecules belonging to the identified library widens the molecular space of the bioactive chemical domains, as required by the subsequent design process to control specific intermolecular interactions within signaling pathways.

## 2. Results and Discussion

### 2.1. RA, RES, SM27, and DOBE Establish Direct Interactions with the Extracellular FGF2/D2 Domain

Nuclear magnetic resonance provided a preliminary screening of the selected library for the interaction with a simplified system formed by FGF2 and the FGFR-D2 domain (hereafter indicated as D2). D2 represents the most relevant domain of the extracellular receptor [34], as it includes the primary interaction site with FGF2 and the positively charged heparin-binding site (Figure 1A). The extracellular portion of FGFR involved in the interaction with FGF2 comprises D2 and D3 immunoglobulin-like domains, together with the D2–D3 linker. However, FGFR-D3 had the characteristics of a molten globule state [18], non-tractable by NMR. Our previous NMR structural investigation further demonstrated that the FGF2/D2 system is a valuable probe to test the activity of inhibitors of FGFR extracellular interaction by showing that the NMR results were in agreement with FGFR activation/inhibition derived from biological tests on endothelial cells [24].

The capability of the studied compounds to bind the FGF2/D2 complex was investigated by acquiring 1D ^1^H NMR spectra of samples containing FGF2/D2 with a fourfold excess of small molecules (FGF2:D2:small molecule 1:1:4). The rationale for selecting this ligand:protein ratio is derived from our previous work on SM27 and RA with the FGF2/D2 system [21,24] (see also Materials and Methods). The changes observed for ligand resonances induced by the presence of the FGF2/D2 complex were analyzed for the five ligands (Figure 2). Chemical shift perturbations were observed for RA, RES, SM27, and DOBE. RES and SM27 resonances were further affected by significant line broadening in the presence of the protein complex, pointing to an exchange of the small ligand between a free and bound state in an intermediate exchange regime on the NMR time scale.

TOL resonances did not change either in chemical shift or in linewidth. Therefore, only RA, RES, SM27, and DOBE were further tested for their biological activity.

### 2.2. RA, RES, SM27 Inhibit FGF2-Induced Endothelial Cell Proliferation

RA, RES, SM27, and DOBE, establishing a direct interaction with the FGF2/D2 complex, were evaluated for their ability to inhibit FGF2-induced endothelial cell proliferation. All the tested compounds, except DOBE, which had only a marginal effect, completely inhibited FGF2-induced BAEC proliferation in a dose-dependent manner (Figure 3A). The compounds presented similar potencies, with mean IG values spanning from 32 µM for RA to 82.7 µM for RES, although without statistical differences (Figure 3B).

To investigate the selectivity of the compounds for the FGF/FGFR system, their activity was also tested on EC proliferation induced by fetal calf serum (FCS) used as a general stimulus, as it contains several growth factors, including FGF2, epidermal growth factor, platelet-derived growth factor, and insulin-like growth factor 1. SM27 and RA showed preferential inhibitory activity for FGF (in the case of SM27, the difference in GI50 between FGF2 and FCS was statistically significant). However, RES did not show preferential activity for the FGF/FGFR system, suggesting that the compound might interact with—and perturb—other growth factors/growth factor receptor systems (Appendix A).

### 2.3. A Diffusion-Ordered NMR Spectroscopy Analysis Highlights the Efficacy of Ligands upon Complex Dissociation

We previously successfully employed NMR diffusion (DOSY) experiments to show that RA was able to destabilize the FGF2/D2 complex, shifting the equilibrium of the FGF2/D2 complex towards the uncomplexed species [24]. DOSY experiments enable the routine measurement of diffusion coefficients to characterize chemical systems in solution. Diffusion parameters can be related directly to an effective hydrodynamic radius of the species in solution, according to the Stoke–Einstein equation, and are thus sensitive to the structural properties of the observed species, such as size and shape [35]. In the case of protein–protein interactions, DOSY experiments can report on complex formation and are a convenient tool to estimate the affinity of interacting molecules [36].

Here, we used DOSY experiments to evaluate the fraction of complexed proteins in solution and determine the capability of the four molecules to perturb and dissociate the FGF2/D2 complex.

The DOSY spectrum recorded for the FGF2:D2 1:1 complex showed a single exchange-averaged diffusion coefficient (Figure 4). Indeed, when the exchange between monomeric and complexed species is fast on both the chemical shift and the diffusion time scale, a single diffusing species is observed at the weighted average of the diffusion coefficients of the free and bound state of the two proteins, and the weighting factors are the relative population of the respective forms [37,38]
*D*_obs_ = χ_D2free_ × *D*_D2_ + χ_Complex_ × *D*_complex_ = χ_D2free_ × *D*_D2free_ + (1 − χ_D2free_) × *D*_complex_

*D*_obs_ = χ_FGF2free_ × *D*_FGF2_ + χ_complex_ × *D*_complex_ = χ_FGF2free_ × *D*_FGF2_ + (1 − χ_FGF2free_) × *D*_complex_

Considering that the diffusion coefficients measured for D2 and FGF2 are comparable within their experimental errors (Pagano et al., 2021), and χ_D2free_ = χ_FGF2free_, the fraction of complexed protein χ_Complex_ can be calculated as:χ_Complex_ = 1 − [(*D*_obs_ − *D*_complex_)/(*D*_D2free_ − *D*_complex_)]

*D*_D2free_ and *D*_obs_ were experimentally derived from DOSY spectra run on the D2:FGF2 1:1 sample and the D2 sample. *D*_complex_ was estimated based on the previously determined model of FGF2/D2 structure in solution [24]. The FGF2/D2 shape can be assimilated to a rod with a length of 60 Å and a radius of 13 Å. Starting from these values, a *D*_complex_ = 9.67 × 10^−11^ m^2^/s was derived from the following equation describing the translational diffusion of a rod-shaped molecule [39]:$$D=\frac{kT}{f}$$
$$f=3\pi \eta L\frac{1}{\ln\left(\frac{L}{r}\right)-0.3}$$

Under our experimental conditions, based on *D* values measured for three different samples, the fraction of the complex was estimated as 0.39 +/− 0.03, allowing for a rough assessment of its *K_a_* through the following equation [40]:$$K_{a}=\frac{\left(1-\chi_{D2}\right)C_{D2}}{\left(C_{D2}-\left(1-\chi_{D2}\right)C_{D2}\right)*\left(C_{D2}-\left(1-\chi_{D2}\right)C_{D2}\right)}$$
where *C_D_*_2_ is the initial D2 concentration. The estimated *K_d_* for the FGF2/D2 system is *K_d_* = 1/*K_a_*~40 μM. DOSY experiments were then run on the FGF2:D2 1:1 complex in the presence of the four ligands (Figure 4), affording the diffusion coefficients reported in Table 1.

A fourfold excess of RA with respect to the complex induced an increase in the diffusion constant close to the figures observed for the unbound proteins (*D*_D2_ = 1.19 ± 0.04 × 10^−10^ m^2^/s), highlighting its ability to induce complex dissociation. The fraction of the complex decreased from 39% to 4% (Table 1). RES, like RA, caused an increase in diffusion corresponding to a reduced population of the complexed species of approximately 12%. At variance, the addition of SM27 and DOBE induced negligible spectral changes close to the experimental error, indicating that their addition to the FGF2/D2 complex did not significantly perturb its molar fraction.

NMR diffusion experiments highlighted different mechanism of action for RA and RES with respect to SM27 and DOBE. The observation that SM27 behaves as an active inhibitor (Figure 3 and Ref. [21]) prompted us to further investigate its molecular mechanisms of action through a residue-level NMR analysis performed on the two separate protein domains. These studies were performed for all the ligands to detect any preferential binding to FGF2 or to its receptor and to identify putative binding sites. Analysis of spectral perturbations on the separate domains is essential when binding is accompanied by complex dissociation. HSQC spectral perturbation induced by RA and RES additions to the preformed complex FGF2/D2 could not be safely attributed to ligand binding or dissociation, as the two phenomena are concurrent.

### 2.4. Preferential Binding of Ligands to FGF2 or D2 

RA, RES, SM27, and DOBE were tested separately for their interaction with ^15^N-FGF2 or ^15^N-D2. Our previous NMR analysis (CSP, relaxation data, and temperature coefficients) on the mechanism of RA interaction indicated that RA binds preferentially to D2 with micromolar affinity, hindering the FGF2/D2 interface and favoring the dissociation of the FGF2/D2 complex [24]. Previously performed SM27 interaction studies with FGF2 revealed a micromolar affinity, whereas no information was available on its interaction with D2 [21].

#### 2.4.1. A Ligand-Based Point of View

Analysis of 1D ^1^H NMR data recorded on FGF2:ligand or D2:ligand at a 1:4 molar ratios supports a preferential binding to D2 for RA and RES. RA and RES resonances exhibited a diffuse line broadening and significant chemical shift changes in the presence of the D2 domain, whereas only minor effects were visible upon FGF2 addition (Figure 5).

Analysis of ^1^H resonances of SM27 and DOBE alone and in the presence of the single domains did not highlight a preferential interaction with one of the two proteins (Appendix A), although SM27 exhibited a slightly higher affinity for both proteins than DOBE, on the basis of the entity of chemical shift perturbation.

#### 2.4.2. A Protein-Based Point of View

NMR HSQC-based experiments on ^15^N-labeled D2 or FGF2 proteins in the absence and presence of 1:4 amounts of the studied ligands allowed us to identify global and residue-specific effects in response to the addition of the library molecules and infer the mechanism of action. Chemical shift perturbations and intensity variations of amide peaks are sensitive to the chemical environment and mobility, respectively, allowing for the identification of preferential binding epitopes and allosteric sites.

The residue level effects of the different ligands on D2 and FGF2 amide resonances are reported in Appendix A and mapped on the structures in Figure 6 and Figure 7. All ligands induce CSP on the D2 receptor domain, but the most significant CSP entity, when compared with that of the other ligands, is due to RA. The comparison of CSP measured for D2 and FGF2 is consistent with the preferential binding of RA to D2, as previously shown [24].

Although of minor entity, the CSP profile induced on D2 by RES addition is reminiscent of that observed for RA addition (Appendix A). The less pronounced changes possibly reflect a lower affinity for the receptor domain. RA and RES binding share a pattern of induced perturbations covering residues: 162–168 (βA and βA’ strands) conserved in all members of the FGFRs family and belonging to the primary binding site of D2 with FGF2; 242–244 (βF strand) at the FGF2/D2 interface; and 214–218 (βD strand) (Figure 6). Both regions represent the direct binding sites of RA RES on D2.

Significant SM27-induced CSP perturbations involve residue K176 (βB strand) in the heparin-binding canyon and Y229, V232, and E234 on the opposite βF strand (Figure 6 and Appendix A). SM27-induced intensity variations involve the C-terminal apical region (residues 169–172 and 251–256) (Figure 6, lower panel, and Appendix A), as well as residues K164 and R168 (heparin-binding canyon). The charged region defined by K164, R168, and K176 likely represents the SM27 binding site on D2. Interestingly, the perturbed C-terminal D2 tail (251–258) is part of the D2–D3 linker, which is allosterically perturbed by RA addition based on NMR amide temperature coefficients [24].

DOBE induced minor CSP perturbation both at the level of residues 166–168 (βA’ strand) and at the level of regions affected by SM27 (K176, N228, and E234) (Figure 6 and Appendix A). The C-terminal D2 tail showed intensity perturbations, as observed for SM27.

Interestingly, residues 166–168 (βA’ strand) at the FGF2/D2 interface were affected by the three active ligands through direct (RA and RES) and allosteric (SM27) perturbations.

FGF2 chemical shift perturbation and intensity variations upon ligand addition were also investigated. SM27induced the most significant CSP and intensity perturbations compared to the other ligands. SM27 affected the chemical shift of the residues that limit the FGF2 heparin-binding canyon (D46, Y124, K128, G131, L135, and I146), thus defining the binding site on the growth factor (Figure 7 and Ref. [21]). Intensity changes were observed for residues T139 and A145, belonging to heparin-binding site R69, located in the allosteric D3 interface [21], as well as L7, T9, L12, E14, G17, and A20 in the N-terminal tail, undergoing a change in the dynamic regime as a consequence of SM27 binding [21].

Although a minor entity, the CSP profile induced by RA addition to FGF2 is reminiscent of that observed for SM27 for residues at the boundary of the heparin-binding site (H44, G47, V49, and G142). Moreover, RA addition perturbs residues located at the FGF2/D3 interface (G24, K27, E67, E68, and G70), a region dynamically affected in the presence of SM27 [21]. This region also showed extensive intensity variations (G4–G24 region of the N-terminal tail, V71 in the βV strand, R81 in the βV-βVI loop, and K154 in the C-terminal segment). Finally, RA addition perturbs residue M151 (CSP) and K154 (intensity), located at the interface with D2.

RES addition perturbed the FGF2 CSP profile at the heparin-binding site (K134 and T139), at the interface with the D3 domain (G70, V72, and F26), and the contiguous C-terminal segment (S152). The observed effect is reminiscent of RA action (Figure 7, upper panel). Intensity variations were instead negligible.

DOBE addition did not significantly affect FGF2 CSP or intensity profiles (Appendix A).

## 3. Materials and Methods

### 3.1. Samples

Unlabeled and ^15^N FGF2 and D2 domains were purchased from ALSA Biotech. The D2 domain is constituted by FGFR2 residues 145–258 (Bmrb id 5943), with a C-terminal His-tag:

MNSNNKRAPYWTNTEKMEKRLHAVPAANTVKFRCPAGGNP MPTMRWLKNGKEFKQEHRIGGYKVRNQHWSLIMESVVPSD KGNYTCVVENEYGSINHTYH LDVVLVPRGSLEHHHHHH

FGF2 is constituted by 155 residues (bmrb id 4091). FGF2 numbering is from amino acid residue 1, as deduced from the cDNA sequence encoding the 155-residue form. In addition, the two surface-exposed cysteines at positions 78 and 96 were mutated to serines, as reported in [21,41].

### 3.2. NMR Experiments

NMR spectra were recorded at 25 °C with a Bruker DMX spectrometer operating at 600 MHz and an Avance II operating at 500 MHz and processed with *TopSpin*^TM^ 4.08 Bruker’s software for NMR acquisition and analysis. NMR experiments on the FGF2/D2 complex were performed with an FGF2:D2 1:1 mixture dissolved in 30 mM deuterated phosphate buffer, pH = 6.5. For the NMR investigation of the changes induced by the addition of inhibitors to protein samples (FGF2/D2 complex, D2, or FGF2), a 0.4 mM protein solution was split into two NMR tubes. The ligand under investigation was added to the first tube, and an identical buffer volume was added to the second tube. The required NMR experiments (1D ^1^H, ^1^H-^15^N HSQC, or DOSY) were recorded subsequently for the apo and holo samples. A fourfold excess of the ligand to the protein was selected to reproduce the experimental conditions employed in our previous works on sm27 and RA [21,24]. These experimental conditions meet the concentration restrictions related to the low solubility of SM27 and RES and enable a comparative study of all the investigated ligands.

DOSY spectra were acquired with stimulated echo pulse sequence, and matrices of 16384 (t2) by 80 points (t1) were collected. The z-axis gradient strength varied linearly from 2% to 98% of its maximum value (53 G cm^−1^), the gradient pulse duration was 4.4 ms, and the time period between the two gradient pulses was optimized on each DOSY spectrum: a range of 170–180 ms was employed for the FGF2/D2 complex in the presence or absence of the ligand, whereas a range of 100–120 ms was employed for the single protein domains with or without ligands. The relaxation delay D1 was set to 1 s. Water suppression was achieved using a 3–9–19 pulse sequence with gradients. Self-diffusion coefficients (*D*) were derived by fitting the NMR data to the Stejskal–Tanner equation [42]. Selected diffusion profiles for the FGF2/D2 complex in the absence and presence of the ligand are shown in Appendix A. The diffusion constants observed for the unbound proteins were *D*_D2_ = 1.19 ± 0.04 × 10^−10^ m^2^/s and *D*_FGF2_ = 1.15 ± 0.04 × 10^−10^ m^2^/s.

^1^H-^15^N-HSQC experiments were acquired using a spectral window of 14 ppm and 2048 complex points in the 1H dimension and a spectral window of 40 ppm and 256 complex points in the ^15^N dimension. A total of 32 transients were acquired for each spectrum. Gradient-tailored excitation was employed for water suppression to reduce signal losses for exchangeable protons.

Chemical shift perturbations were calculated as a weighted sum of ^1^H and ^15^N chemical shift changes [43], and the standard deviation calculated across the whole protein sequence was used as a threshold value to map significant perturbations. Peak intensities in HSQC spectra were measured by Sparky software (University of California, San Francisco). The intensities of peaks belonging to each spectrum were normalized by dividing their intensity by the total intensity, that is the sum of all the measured cross peaks.

PyMol software (DeLano Sci LLC, San Carlos, CA 94070 USA) was employed for graphical representations. For graphical representation of the NMR results, the FGF2 NMR structure (PDB ID: 1BLD) was employed and superimposed to chain A of the FGF2/D2D3 X-ray dimeric structure (PDB ID: 1EV2) in contact with chain E, including the D2 and D3 domains.

### 3.3. Endothelial Cell Proliferation Assay

Bovine aortic endothelial cells (BAECs) provided by E. Dejana (Milano, Italy) were cultured in DMEM with 10% FCS. BAECs (2500 cells/well) were seeded into 96-well culture plates in DMEM 1.5% FCS. After 24 h, the medium was substituted with DMEM containing FGF2 (10 ng/mL) and 1% FCS or DMEM with 3.5% FCS, with the indicated concentration of the compounds. After 72 h, cells were fixed and stained with crystal violet solution (Sigma–Aldrich). The staining was eluted with a 1:1 ethanol/0.1M sodium citrate solution, and the absorbance was measured at 595 nm. Differences in GI50 values (the concentration of compound causing 50% inhibition of cell proliferation) were analyzed by two-way ANOVA followed by Tukey’s multiple comparison test, using GraphPad Prism 8 (GraphPad, La Jolla, CA, USA).

## 4. Conclusions

We demonstrated by NMR that four low-molecular-weight ligands (RA, RES, SM27, and DOBE) establish specific interactions with the extracellular portion of the FGF2/FGFR system. NMR diffusion experiments exploited to investigate the capability of each ligand to disrupt FGF2/D2 assembly indicated that only RA and RES dissociate the complex.

Residue-level NMR interaction studies of the ligands with the FGF2/D2 complex and with the single proteins, namely FGF2 and D2, indicate that the ligands exert distinct mechanisms of action. RA and RES preferentially bind to D2 at the level of its interface with FGF2, competing for the same binding site and thus hindering FGF2/FGFR complex formation. SM27 preferentially engages the FGF2 surface at the level of residues located in the long b10-b12 loop, which is part of the reported heparin-binding site. Touching the heparin-binding region produces effects on an FGF2 distal region at the interface with D3, as previously shown [21], thus explaining why SM27 does not dissociate the FGF2/D2 complex. The mechanisms highlighted by the NMR molecular studies indicate that the differing effects of the inhibitors on complex stability are related to their preferential binding site. The finding that RA, RES, and SM27 were all active in inhibiting FGF2-induced cell proliferation with comparable potencies indicates that both mechanisms of action are at work in inhibiting the biological effects elicited by the FGF/FGFR complex. Despite the preferential binding to one of the two partners, RA, RES, and SM27 engage direct and long-range perturbations on the other protein domain. DOBE at variance was unable to induce spectral perturbations on FGF2. The marginal effect of DOBE in proliferation assay indicates that its interactions with D2, deduced from NMR analysis, may not be sufficiently strong to translate into functional activity. We believe that the concurrent effect of the ligands on both domains is the basis of an effective inhibitory action.

The knowledge of the molecular mechanisms driving the interactions of potential inhibitors of a selected protein–protein complex (FGF2/FGFR) provided by NMR analysis represents a fundamental step toward establishing effective strategies to control specific intermolecular interactions within the studied signaling pathways.

## Data Availability

Not applicable.

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
