# Peer review of "Inhibition of FGFR Signaling by Targeting FGF/FGFR Extracellular Interactions: Towards the Comprehension of the Molecular Mechanism through NMR Approaches"

_ijms, 2022, doi:10.3390/ijms231810860_

Round 1
Reviewer 1 Report
The idea uses translational mobility to study the interaction substrate to proteins and their complexes not new, but not utilized often. I agree that the authors presented interesting experimental results obtained by diffusion NMR techniques. Nevertheless, I suppose that NMR experiments based on transverse relaxation data deliver data provided for more precise analysis. As I understand, the current studies are continuation projects performed on RA, as described in the previous paper (Pagano et al. (2021) ChemBioChem, 22, 160-169). In the manuscript, the authors include four additional ligands but limit their studies primarily to translational diffusion.
In my opinion, the accuracy of the evaluated diffusion coefficient is rather overfitted. The values presented in Table 1 (page 9) contain five digits after a point (without errors). Do they expect such accuracy? Are the gradient units of Bruker DMX providing such stability for gradient pulses? The heating element regulated temperature is usually mounted in the down path of the NMR probehead. If so, a temperature gradient in your NMR sample decreases accuracy. In previous paper, the diffusion coefficient for FGF2 and FGFR-D2 were calculated as 1.14 +/- 0.045 x10-10 m2/s, and 1.18 +/- 0.039 x10-10 m2/s, respectively.
An additional question about Figure 4. As the caption follows, the diffusion data for the FGF2/D2 (black ones) have to be the same on all four panels. The differences appear in the presence of the library molecules (red). But in my opinion, the experimental data presented as black are different.
Finally, a lot of references are missing in the text. For instance, lines 70 – 71 ‘… heparan sulfate proteoglycans, as demonstrated by SPR and cell-based binding assays.’ Reference should be here. A similar situation in a couple of other places in the manuscript. Authors have to go through the whole manuscript to include references, where required. The pdb code used for structural analysis is not provided in the Materials and Methods section. Probably, there are 1BLD and 1EV2 (based on the previous paper).
I’m not talking about literature, which has to be corrected. This will be done during the preparation of the final version of the manuscript.
Reviewer 2 Report
The manuscript is an interesting study on FGFR to provide a new strategy to affect its function. The study has not well presented and I hope authors can make the conclusion clear and solid.
Major,
1. it will be good to add the domain structure of FGFR in Figure 1 to let readers understand the location of the ATP binding pocket and other regions.
2. the 1D NMR study in figure 2 is quite interesting, it would be good to show the concentration dependent line broadening, eg complex:compound ratio at 10:1, 5:1, 2:1, 1:1 and 1:4. the data does provide specific binding binding mode.
3. CSP data in figure 6 is important, it would be important to show the overlay of the spectra. what is the binding affinity based on the NMR studies?
4. the current manuscript does not show whether complex is critical for binding, please re-consider about figure 5 and figure 6. I think 2D HSQC experiment can give a clear idea of requirement of the complex.
5. for the diffusion experiment, it is unclear that the coefficient shown in the table. Several D should be shown, D for free D2, FGF2, D2/FGF2 and D2/FGF2 + compound, then the conclusion can be made.
Minor
1. please provide the PDB code in figure 1.
2. line 113, please remove the symbol at the beginning of the sentence.
3. the IC50 measurement is quite interesting, is the assay specific for measuring FGFR pathway? Based on the method, it does not look to be an IC50 assay. it is more like a GC50 assay and it may not be concluded that these compounds target FGFR. please clarify.
4. please change the titles from questions to a statement.
Reviewer 3 Report
Pagano and co-workers produced an excellent NMR-based study to characterise the interactions between FGF2, FGFR D2 domain (D2) and small molecule modulators. The use of simple 1H-based line broadening experiments (at low or equal protein-to-ligand ratio) is pleasing to see, as it is a simple and quick approach to identify strong binding ligands. It is something that the NMR community is not doing enough of in my opinion. The experiments clearly illustrated the binding of the ligands. I think the authors could do a few titration (of the protein(s)) experiments to work out the binding affinity of the ligands when both FGF2 and D2 were present (e.g., Figure 2), or when only one of them was present (e.g., Figure 5), which I think would be useful to the readers. It would also be useful if the authors could include the protein and ligand concentrations in the figure legends. The diffusion experiments are interesting. Our laboratory has also used DOSY to look at protein complexes and we typically integrate the whole protein background but avoiding areas with the ligand signals. It would be useful if the authors could detail the areas that they used for the integration, and perhaps show the diffusion curves in the Supplementary Information. For the protein NMR part, some of the chemical shift perturbations observed are really small. I wonder if the authors could show us a few spectral overlays so that we could see if these are ‘real’ chemical shift differences or not. For experimental, please include the relaxation delay, especially for the DOSY experiment, which is very important.
Round 2
Reviewer 2 Report
The revision looks good for publication.
Reviewer 3 Report
Thank you for providing the revised manuscript, which is much clearer in my opinion. I am happy to recommend its publication.
PS: Please note that the relaxation delay in the DOSY experiments that I was referring to was the "D1" value in Bruker TopSpin.